# Learning Reusable Manipulation Strategies

**Jiayuan Mao    Joshua B. Tenenbaum    Tomás Lozano-Pérez    Leslie Pack Kaelbling**
Massachusetts Institute of Technology

**Abstract:** Humans demonstrate an impressive ability to acquire and generalize manipulation "tricks." Even from a single demonstration, such as using soup ladles to reach for distant objects, we can apply this skill to new scenarios involving different object positions, sizes, and categories (e.g., forks and hammers). Additionally, we can flexibly combine various skills to devise long-term plans. In this paper, we present a framework that enables machines to acquire such manipulation skills, referred to as "*mechanisms*," through a single demonstration and self-play. Our key insight lies in interpreting each demonstration as a sequence of changes in robot-object and object-object contact modes, which provides a scaffold for learning detailed samplers for continuous parameters. These learned mechanisms and samplers can be seamlessly integrated into standard task and motion planners, enabling their compositional use.

**Keywords:** Contact Modeling and Manipulation, Task and Motion Planning

## 1   Introduction

Humans possess an exceptional ability to acquire and generalize manipulation "strategies." Even from a single demonstration of using a soup ladle to reach a distant object (Fig. 1a), we can generalize and reuse this strategy in various novel scenarios: to different object positions and sizes, and even to diverse object categories like forks and hammers. Furthermore, humans can combine these strategies to devise long-term plans, perhaps using a soup ladle to hook an apple, place it into a bag, and move the bag to a shelf, dramatically expanding the scope of our manipulation abilities.

A salient feature of these manipulation tactics is that they can be expressed as a sequence of *basis manipulation operations*, characterized by varying contact modes between robots and objects. For example, as illustrated in Fig. 1b and c, the "hook-using" strategy comprises a series of four contact modes: the free movement of the arm, tool grasping while applying contact force between the tool and the target, tool placement, and ultimately, target grasping. The continuous parameters of these operations, in principle, can be produced by generic samplers and motion planners, but planning in terms of these generic basis operations can be very slow due to a long planning horizon with substantial branching due to choice of basis operations and continuous parameters.

To tackle these challenges, this paper presents a framework that equips machines with the ability to *learn, generalize, and reuse* such manipulation strategies, referred to as "*mechanisms*," through a single demonstration and subsequent self-play in a distribution of target problems. The key insight driving our framework is the characterization of each mechanism as a sequence of contact mode changes between the robot and objects, complemented by a specialized sampler that generates grasps, contacts, and trajectories tailored specifically for the mechanism. Our framework takes an explanation-based learning approach [1, 2], departing from conventional methods that learn policies or parameterized trajectories from large numbers of demonstrations. In particular, our framework extracts an abstract representation that explains the underlying contact interactions between objects from the demonstration, then during the self-play stage of mechanism learning, the agent explores feasible actions that align with the demonstrated contact mode sequence, generalizing to different objects and initial configurations. Leveraging successful trials from self-play, we train samplers

---

Project page: https://concepts-ai.com/p/mechanisms

7th Conference on Robot Learning (CoRL 2023), Atlanta, USA.

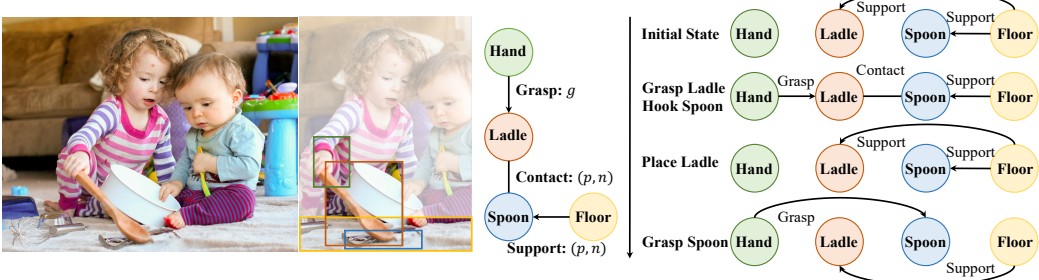

**(a)** Demonstration of using a soup ladle to reach for the spoon

**(b)** The contact mode graph between objects, with continuous parameters.

**(c)** A sequence of contact mode graphs in the process of using the soup ladle to reach for the spoon.

Figure 1: A single demonstration can teach a reusable multi-step manipulation strategy.

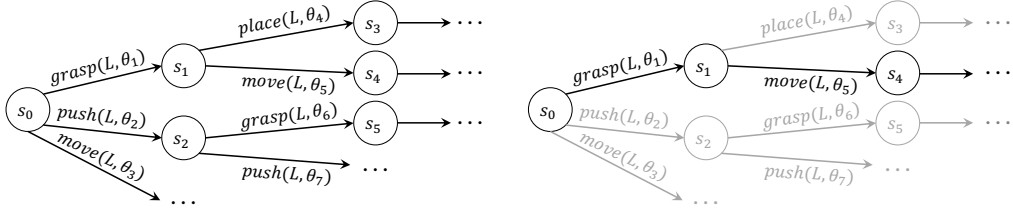

(a) Search in the large hybrid space of discrete mode families and continuous motion parameters.

(b) Search in the hybrid space guided by the mechanism. Only continuous parameters needs to be searched.

Figure 2: (a) Planning with basis manipulation operations usually involves a search in the hybrid space of discrete contact mode families (free, holding, pushing, etc.) and continuous parameters (grasping pose, moving trajectories, etc.). (b) Learning mechanisms can help reduce the search space by specifying a sequence of discrete transitions between contact mode families.

that are tuned specifically for each mechanism. The learned mechanisms and samplers can then be recombined to efficiently devise long-horizon plans for novel goals in novel environments, by reducing the effective search horizon and focusing the sampling process during planning.

This formulation introduces a significant capacity for generalization via abstraction and compositionality: by extracting the contact-mode sequence from the demonstration, we retain its most causally important aspects while abstracting away many irrelevant details, and by representing learned mechanisms in a form similar to basis operations, we are able to leverage general-purpose task-and-motion planners to obtain a truly compositional system. The contributions of this paper are: a novel representation of complex manipulation actions in terms of mechanisms, an algorithm for learning new mechanisms from a single demonstration and subsequent self-play, and a planning framework for integrating new mechanisms with other manipulation primitives, including those that are briefly dynamic [3], to solve novel problems.

## 2 Planning with Contacts and Mechanisms

Our framework is based on hybrid task and motion planning. We begin with a small set of generic *basis manipulation operations* corresponding to different *contact mode families* [4] between robot and objects (Table 1). Sequences of these basis operations form a rich class of manipulation strategies, such as using soup ladles to reach for distant objects, or pushing plates to the table edge to pick them up. In theory, a complete search algorithm could discover such strategies and find sequences of them to solve difficult novel problems, without any demonstrations at all, as illustrated in Fig. 2a. However, planning at this level is inefficient to the point of infeasibility. Therefore, in this paper, we will explore learning "macros" of basis operations, using a compatible representation that will allow learned mechanisms to, themselves, be composed to solve harder problems.

### 2.1 Basic Domain Representation

We adopt a representation similar to those used in the task and motion planning (TAMP) literature [5]. Formally, given a space $\mathcal{S}$ of world states, a TAMP problem can be defined as a tuple $\langle \mathcal{S}, s_0, \mathcal{G}, \mathcal{A}, \mathcal{T} \rangle$. Here, $s_0 \in \mathcal{S}$ denotes the initial state, which includes the shapes and poses of all objects. $\mathcal{G} \subseteq \mathcal{S}$

| Name | Args. | Description |
|------|-------|-------------|
| **transit** | ⟨*empty*⟩ | Move the robot without object collisions. |
| **transit-cont** | *?x, ?s* | Move without holding anything, but have a contact with *?x*. *?x* is supported by *?s*. |
| **grasp** | *?x, ?s* | Grasp the object *?x* that is currently supported by *?s*. |
| **place** | *?x, ?s* | Place object *?x* onto *?s*. *?x* should be currently held by the robot. |
| **move** | *?x* | Move while holding *?x*. |
| **move-cont** | *?x, ?y, ?s* | Move while holding *?x* and *?x* have contact with *?y*. *?y* is supported by *?s*. |

Table 1: The list of basis operations. Continuous parameters are omitted. "cont" refers to "contact."

```
(:action move-with-contact
 :parameters (?tool - item ?target - item ?support - item
              ?param - contact-move-param ?qt - trajectory)
 :precondition (and
   (holding ?tool)
   (support ?target ?support)
   (valid-contact-move-param      ... ?param) ;; points+normal
   (valid-contact-move-trajectory ... ?param ?qt))
 :effect (and
   (assign (robot-config) ...)  ;; update robot position
   (assign (pose ?tool)   ...)  ;; update tool position
   (assign (pose ?target) ...)  ;; update target position
```

```
(:mechanism hook
 :parameters (?tool - item ?target - item ?support - item ...)
 :precondition (and ;; the initial contact mode graph
   (handsfree) (support ?tool ?support) (support ?target ?support))
 :certified (and      ;; the goal of the mechanism
   (support ?tool   ?support)
   (holding ?target))
 :actions (ordered   ;; the sequence of contact primitives
   (grasp     ?tool ?support)
   (move-cont ?tool ?target ?support)
   (place     ?tool ?support)
   (grasp     ?target ?support)))
```

**(a)** The modeling of the robot basis operation *move-with-contact* using a STRIPS-like syntax..

**(b)** The precondition (initial contact modes), goal (target contact modes), and the sequence of actions for a mechanism.

Figure 3: The syntax used to model basis operations and mechanisms.

represents the goal specification, often expressed as a logical expression evaluated based on a state (e.g., *holding*(Spoon)). $\mathcal{A}$ denotes a set of continuously parameterized actions that the agent can execute, such as grasping and placing objects. Finally, $\mathcal{T}$ is a partial environmental transition model defined as $\mathcal{T} : \mathcal{S} \times \mathcal{A} \to \mathcal{S}$. Each action $a$ is parameterized by two functions: the precondition function $pre_a$ and the effect function $eff_a$. The semantics of these functions is that for any state $s \in \mathcal{S}$ and action $a \in \mathcal{A}$, if $pre_a(s)$ holds, then $\mathcal{T}(s, a) = eff_a(s)$.

**State representation.** An environmental state is represented as a tuple $s = (\mathcal{U}_s, \mathcal{P}_s)$. $\mathcal{U}_s$ denotes a set of objects, including the robot and other physical objects, and we assume it is fixed during the execution of actions. Objects in $\mathcal{U}_s$ will be referred to using names such as *SoupLadle* and *Floor*. The set $\mathcal{P}_s$ contains *state variables* (atoms in STRIPS). Each state variable contains a predicate name (e.g., *pose*), a list of object arguments (e.g., *SoupLadle*), and a value (e.g., the pose of the soup ladle in *SE*(3)). In addition to shapes and pose variables, $\mathcal{P}_s$ also contains two sets of variables to represent the contact mode graph: *holding*(*?x*) and *support*(*?x, ?y*), where *?x* and *?y* are instantiated with all objects in $\mathcal{U}_s$. These variables describe whether the robot is holding an object *?x* and whether the object *?x* is supported by *?y*, respectively[*].

**Basis operators.** The basis operators are specified as parameterized *operator schemas*, ⟨*name, args, precond, effect, sampler*⟩, where *name* is the name of the schema, *args* is a list of arguments, including both object arguments and continuous parameters. These continuous values can be generated by invoking the *sampler*, possibly conditioned on other aspects of the state, such as the shapes of the objects involved. The precondition *precond* and effect *effect* are logical expressions over variables in *args* and will be evaluated at the current state. A schema can be *grounded* into a concrete basis operation $a$ by specifying its arguments. See Figure Fig. 3a for a concrete example and Appendix A.3 for more discussion.

## 2.2 Mechanisms

A manipulation *mechanism* is defined as a sequence of basis operations with a specialized sampler, in order to accelerate planning with generic basis operations. Formally, each mechanism is represented as a tuple of ⟨*args, precond, certified, actions, sampler*⟩. *args* is a set of arguments. *precond* is the initial contact mode graph including *holding* and *support* relations. *certified* is the goal of the mechanism, which usually specifies the final contact mode graph. *actions* is an ordered list of primitive operations. *sampler* is the specialized sampler that can generate continuous parameters for all basis operations in *actions*. Fig. 3b illustrates the definition of the "hook-use" mechanism. In this case, three objects are involved: the tool object (e.g., the soup ladle), the target object (e.g., the

---

[*]Following the STRIPS convention, we will be using names such as *?x* and *?y* for variables and strings such as *SoupLadle* for objects.

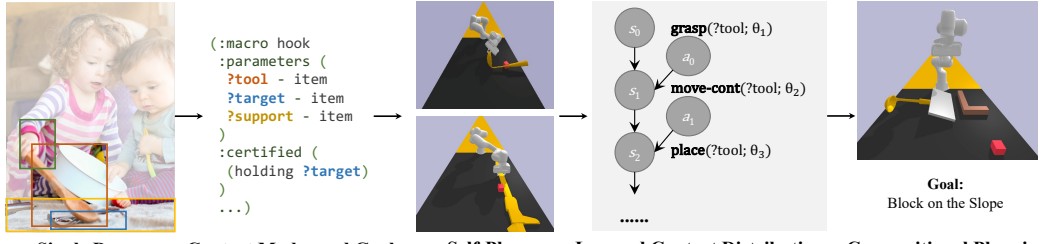

| Single Demo | Contact Modes and Goals | Self-Play | Learned Contact Distributions | Compositional Planning |

Figure 4: Learning process: extract contact modes and goals from demonstration; train specialized sampler via trial-and-error in new situations; add new mechanism to planner.

spoon), and the support object (e.g., the floor). The goal of the mechanism is to grasp the target object that was initially out of reach. This macro contains a sequence of four basis operations: grasping the tool from the support, moving while holding the tool and "indirectly" pushing the target object, placing the tool back to the support, and finally grasping the target object. It also has an associated sampler that generates feasible grasps of the tool and contacts between the tool and the target object.

## 2.3  Planning with Basis Operators and Mechanisms

To plan using the basis operations, mechanisms, and samplers, we employ a simple bilevel search approach, similar to the adaptive algorithm in PDDLStream [6]. Illustrated in Algorithm 2 in Appendix A.1, we first apply symbolic STRIPS search algorithms with the fast-forward heuristic [7] to explore discrete action plans (i.e., candidate sequences of basis operations and objects to move). Subsequently, we use samplers to find suitable continuous parameters. We iteratively call the samplers associated with each operation to generate continuous parameters, such as grasps, contact surfaces, and trajectories. We simulate the grounded operators to verify their effects, and backtrack to a new discrete plan if continuous parameters cannot be found. Mechanisms can be directly integrated into the bi-level search-based planner as additional operators, with their own specialized samplers. Intuitively, as illustrated in Fig. 2b, incorporating mechanisms in search prunes out the branching factor of possible contact modes and also improve efficiency in sampling continuous parameters.

The use of a physical simulator to verify action effects enables us to handle *briefly-dynamic* tasks, in which the robot controllers are position-based, but may initiate situations in which the objects experience acceleration and velocity before they reach a stable configuration. Examples include objects sliding down inclines, or tipping upward when weights are placed on them (see Appendix A.2).

## 3  Learning New Mechanisms

Our goal is to learn a new *mechanism* from a single demonstration and a distribution of target problems. The demonstration includes a sequence of robot actions and object contacts, as well as a human-specified goal, which will be the *certified* effect in the new mechanism definition. We assume access to an environment simulator that can generate random initial configurations of objects such that the target mechanism is applicable. For example, in the *hook-use* case, the environment contains two objects of various categories placed on the table so that one object is within reach and the other object is out of reach. The overall framework is depicted in Fig. 4: The learning algorithm first extracts the contact-mode changes in the demonstration, resulting in a sequence of basis operations. Next, it generates self-play trajectories that align with the basis-operation sequence but now with novel objects in novel initial configurations (e.g., using spoons to reach for forks). This self-play step involves trial-and-error interaction with the environment. Based on the successful trajectories, we learn a sampler that generates mechanism-specific contacts between objects (e.g., grasp of tools in order to reach for distant objects), and finally add it to our repertoire of planning operators.

### 3.1  Extraction of Preconditions and Operation Sequence

We first segment the demonstration trajectory based on robot-object contacts: free, holding, and push motion. Each segment will correspond to one step in the mechanism macro. Next, for each segment, we build a contact graph between the robot hand, the object that is in contact with the robot (including

---

**Algorithm 1** Sampler Learning Algorithm

---

1: **procedure** SAMPLERLEARNING($env, m$)
2:     Initialize classifiers $f_i$ and dataset $\mathcal{D}_i \leftarrow \emptyset$ for each continuous parameters $i$ in $m$.
3:     **for** each iteration **do**
4:         $s_0 \sim env.reset()$                                                   ▷ Sample an initial configuration
5:         $gm \leftarrow$ RandomGrounding($m$)          ▷ Apply the mechanism on a random set of objects
6:         $plan \leftarrow$ CONTINUOUSSEARCH($s_0, gm.certified, gm.actions$)
7:         **for** each continuous parameter $i$ **do**
8:             **for all** $\theta$ sampled in CONTINUOUSSEARCH for parameter $i$ **do**
9:                 **if** $\theta$ is in $plan$ **then** Add $(\theta, 1)$ to $\mathcal{D}_i$                        ▷ Successful samples
10:                **else** Add $(\theta, 0)$ to $\mathcal{D}_i$                                           ▷ Failed samples
11:            Update classifier $f_i$ using $L$ and $\mathcal{D}_i$

---

holding or pushing, if any), and the object that is in contact with the held object ("indirect contact"). Finally, we add all the objects that support the objects (exerting forces that point in the $+z$ direction). By completing the relationships among objects, we obtain the sequence of basis operations. The precondition of the mechanism operator corresponds to the initial contact modes. After extracting the initial contact modes and the basis operation sequence that are grounded on concrete objects in the demonstration (the soup ladle, the spoon, and the floor), we *lift* them into an abstract mechanism definition by replacing concrete object names with variables, as illustrated in Fig. 3b.

### 3.2 Sampler Learning

The sampler learning process is an iterative exploration within a simulated environment. Algorithm 1 describes the high-level process of learning a sampler based on the success of generated samples in achieving the goal. In each iteration, we randomly sample an initial configuration from the simulator and attempt to execute the mechanism on objects present in the environment. Notably, instead of searching for a plan to accomplish the mechanism's goal with all available basis operators, we require the search algorithm to adhere to the sequence of basis operations derived from the demonstration. Specifically, we fix the discrete-level plan and employ generic samplers for grasping, contact, and trajectory generation to locate plans that satisfy the goal. All continuous parameters sampled during the search will be labeled as "1" or "0" based on whether they present in the successful plan. Note that even if the environment does not contain distractors and the mechanism is always applicable, we still need to explore different choices of arguments (e.g., use which object to reach for the other one).

Leveraging this labeled dataset of samples, we can train an additional sampler that generates mechanism-specific samples of continuous parameters. For example, in a scenario where a soup ladle is used to reach distant objects, it is favorable to grasp the handle—a more mechanism-specific action, rather than generic grasps. This can be formalized as learning a distribution of "successful" continuous parameters for actions within a mechanism. We do this by training a score function to rank samples produced by the generic samplers[†]. In essence, for each continuous parameter in the mechanism (grasping poses, contact surfaces, and trajectories), given a dataset of samples and their success labels $\mathcal{D} = (\theta, label)$, we train a classifier $f$ that estimates a scalar value in $[0, 1]$ representing the probability that $\theta$ can result in a successful application of the mechanism, using binary cross-entropy loss: During deployment, we use the generic sampler to generate a batch of samples, which we sort based on the predicted likelihood of success.

Note that although we train classifiers for individual parameters, these classifiers are designed to be conditioned on previously generated parameters as defined in the basis operation schema. Hence, their sequential application is capable of generating a joint distribution of successful parameters for mechanism applications in an "auto-regressive" manner. The benefit of this factorized approach, as opposed to learning a single classifier over the concatenation of all parameters, is its efficiency during the search process. If the execution of one step in the mechanism fails, the search algorithm can immediately backtrack, thereby averting wasteful continuation into subsequent steps.

---

[†]More sophisticated generative models will work, especially when the samples are hard to generate.

### 3.3 Implementation

Our algorithm for learning mechanisms is general purpose. In order to demonstrate its value, we have implemented a concrete system with the basis operators in Table 1.

**Samplers for Basis Operators** There are three types of continuous variables to be sampled for the basis operators described in Table 1: grasping poses relative to an object (represented as $SE(3)$ poses of the end-effector), placement poses (represented as $SE(3)$ poses in the support object), contacts between two objects (represented as the $SE(3)$ pose of object 1 in the frame of object 2), and robot arm trajectories (represented as a sequence of joint configurations). The samplers are designed to be generic, relying solely on geometry and not specific object semantics (e.g., soup ladle grasping). We include one here as an example and describe the full list in Appendix A.4.

***Object Contact** ($C$).* For both robot-object and object-object contact, the object contact sampler, $C(\mathcal{O}_1, \mathcal{O}_2, T_{o2}, \mathcal{O}_s, T_s)$ takes five arguments, including the current holding object $\mathcal{O}_1$ (or the robot gripper itself when not holding anything), the object to contact $\mathcal{O}_2$ and its current pose $T_{o2}$, and the object that supports $\mathcal{O}_2$: $\mathcal{O}_s$ and its pose $T_s$. It first randomly samples two surfaces, represented as a pair of (point, unit normal), $(p_1, n_1)$ and $(p_2, n_2)$ on $\mathcal{O}_1$ and $\mathcal{O}_2$ respectively. Since we do not consider pushing $\mathcal{O}_2$ "towards" the supporting object $\mathcal{O}_s$, we additionally require that $n_2$ is perpendicular to $n_s$, which is the direction of the support force from $\mathcal{O}_s$ to $\mathcal{O}_2$. Next, it solves for a transform $T$ on $\mathcal{O}_1$ such that $Tp_1 = T_{o2}p_2$ and $Tn_1 = -T_{o2}n_2$ (essentially place $p_1$ on $\mathcal{O}_1$ at $p_2$ and pointing towards $n_2$ to excert force).

**Learned Samplers** Each mechanism has specialized samplers for all of its continuous parameter, organized into a DAG, in which each parameter is sampled conditioned on its parents in the graph (a.k.a. previously sampled parameters). Each sampler has a scoring classifier that processes the target parameter along with any values it is conditioned on and predicts a success likelihood in $[0, 1]$. We employ various encoders for different parameter types. For object shapes, we use a PointNet++ encoder [8] to process the point clouds. For poses, including a 3D translation and a 4D quaternion, we utilize Multi-Layer Perceptrons (MLPs). For contact information, we encode the contact points and normals using MLPs as well. Each of these encoders processes their corresponding inputs into a fixed-length embedding, and the embeddings are then concatenated into a single vector. Finally, we apply a linear transformation followed by a sigmoid activation to output the classification result. We train each classifier on 100 samples. For trajectory-typed parameters, we only apply classifiers to object-object contact trajectories encoded as the contact normal direction and the distance. We do not consider encoding and classifying the actual robot arm trajectory for grasping and placement motions, although it is, in principle, possible to encode them using appropriate encoders.

## 4 Experiments

We conducted experiments in the PyBullet simulation environment, to evaluate the ability of our system to learn individual mechanisms and its ability to reuse learned mechanisms in novel tasks. We describe real-world deployment of the system in Appendix B.3 and more videos on our website.

### 4.1 Learning Mechanisms from Single Demonstrations

**Setup.** Our evaluation encompasses six distinct mechanisms, grouped into two categories: the first four tasks assess "tool-use," including *(Edge)* pushing objects to the edge of a table for pickup, *(Hook)* using tools to reach for distant objects, *(Lever)* flipping objects using heavy objects as levers, and *(Poking)* using tools to poke objects out of a tunnel. The remaining two tasks fall under the "reasoning about stability" category, including *(Center-of-Mass)* achieving stable object placement on another object, *(Slope-and-Blocker)* using objects as blockers to prevent objects from falling off inclined surfaces (illustrated in Fig. 5). The object models are blocks, bricks, bowls, plates, documents, spoons, forks, soup ladles, hairbrushes, hammers, and calipers. The demonstrations are created by executing a manually written script in one specific initial configuration.

During training and testing, each method has access to a distribution of initial configurations and goals. Each task consists of a randomly sampled initial configuration that includes target objects placed on the table and a specific goal to be achieved (e.g., holding one of the target objects). We

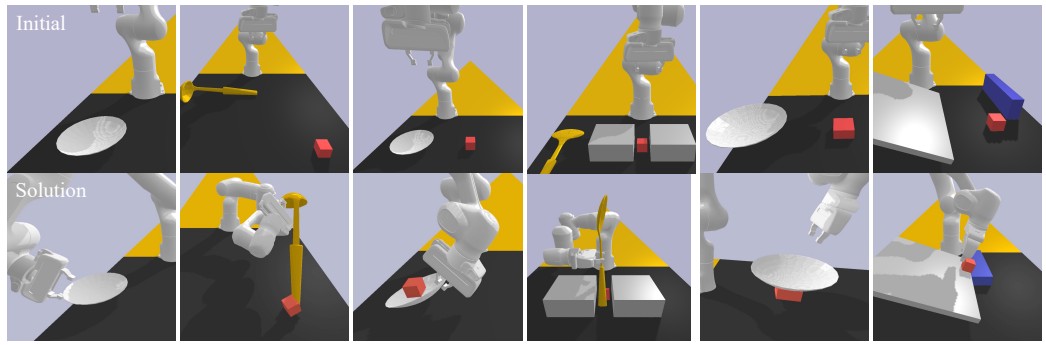

| **Edge:** hold(P) | **Hook:** hold(B) | **Lever:** hold(P) | **Poke:** hold(B) | **CoM:** support(P, B) | **Slope:** support(B, S) |

Figure 5: Illustration of six mechanisms learned by our algorithm. Top row: initial configuration. Bottom row: solution found by our sampling-based planner. P: Plate. B: Block. S: Slope.

| Method | Edge | Hook | Lever | Poking | CoM | Slope&Blocker |
|---|---|---|---|---|---|---|
| Basis Ops Only | $89.45_{\pm 5.53}$ | $>600$ | $523.18_{\pm 9.22}$ | $>600$ | $19.30_{\pm 2.82}$ | $>600$ |
| Ours (Macro) | $8.34_{\pm 2.57}$ | $30.82_{\pm 5.78}$ | $\mathbf{1.38}_{\pm 0.31}$ | $494.30_{\pm 50.01}$ | $17.58_{\pm 1.27}$ | $148.57_{\pm 10.30}$ |
| Ours (Macro+Sampler) | $\mathbf{0.57}_{\pm 0.05}$ | $\mathbf{3.84}_{\pm 1.56}$ | $1.55_{\pm 0.29}$ | $\mathbf{97.76}_{\pm 10.67}$ | $\mathbf{0.97}_{\pm 0.09}$ | $\mathbf{4.11}_{\pm 0.94}$ |

Table 2: Average solution time and stderr of 10 trials, All methods have a 10 mins timeout.

design the distribution of initial object placements to ensure the feasibility of the corresponding mechanism. For example, in the "hook-use" mechanism, we randomly place two objects on the table. Object 1, which is within the robot's reach, is selected from the following categories: soup ladle, hammer, spoon, and hairbrush. Object 2 is a box that is initially placed outside of the robot's reach. We provide more environmental details in Appendix B.1.

**Results.** Table 2 summarizes the results of our experiments comparing our method to planning with the basis operators only, as well as to an ablation of our method in which we learn the basis operation sequence for the mechanism but do not learn specialized samplers. Due to the inclusion of novel object instances and categories in the test environments, simpler baselines, such as replay of the demonstration trajectory, have zero success rate, and are not included. The table shows the average time needed for each planning algorithm. Our method effectively learns mechanisms for all tasks. The macro-only version occasionally timed out, emphasizing the importance of both sequence and sampler learning. As an example, consider the Lever task, in which the robot needs to position an object on a plate to flip the plate up. Having prior knowledge of this strategy at the abstract basis operation level significantly simplifies the task. The learning of a mechanism-specific sampler, in this case, does not offer much additional benefit. On the contrary, task CoM presents a different scenario: here, the robot is tasked with learning a distribution for stably placing an object on a small block, which benefits greatly from learning a specialized sampler. The poking task is difficult, even for learned samplers because it involves a arm motion planning problem with a very small collision-free region. We discuss more details on the importance of sampler learning in Appendix B.2.

### 4.2 Planning with Learned Mechanisms

**Setup.** To test the compositionality of mechanisms, we evaluate different algorithms on two novel complex tasks, illustrated in Fig. 6. **(Push-then-Pick-then-Hook)** the agent needs to utilize a thin caliper, placed on the table to reach for a distant block. However, the caliper must first be pushed to the table side to enable a successful grasp. **(Hook-then-Place-on-Slope)** the agent needs to use a soup ladle as a tool to reach for a distant object. Subsequently, the agent must use either the soup ladle or a brown brick to act as a blocker, preventing the object from falling off a slope. There is no additional training—we used mechanisms learned during the previous experiment. We design the test distribution of object placements to ensure tasks being feasible.

**Results.** Table 3 presents the planning time for all the evaluated methods. In scenarios with numerous objects available for interaction, searching directly for low-level manipulation primitives without the guidance of useful mechanisms can be extremely slow. However, using mechanisms as "macros" in

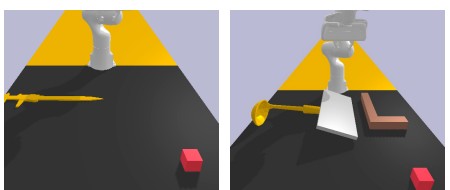

**Task 1:** hold(Block)  **Task 2:** support(Block, Slope)
Figure 6: Illustration of compositional tasks.

| Method | Task 1 | Task 2 |
|---|---|---|
| Basis Ops Only | 0% | 0% |
| Ours (Macro) | 20% | 0% |
| Ours (Macro + Sampler) | 100% | 90% |

Table 3: Average success rate on novel compositional tasks in 10 runs. Timeout is 10 minutes.

the search process significantly enhances efficiency, enabling the system to solve a greater number of tasks within the given time limit. The learned samplers further improve the overall efficiency.

## 5 Related Work

Approaches toward manipulation with primitives have been extensively studied in the field of robotics. These approaches aim to design algorithms that can perform complex tasks by decomposing manipulation tasks into a sequence of primitive actions, such as grasping, pushing, and placing. Task and motion planning algorithms provide methods for generating high-level task plans while considering the robot's physical constraints and the environment's dynamics. These approaches can be roughly grouped into two groups: sampling-based such as HPN [9], IDTMP [10], and many others [11, 12, 6], and global optimization-based such as Logic-Geometric Programming [13] and Pang et al. [14]. Our method falls into the group of sampling-based planning, and we extend existing approaches to handle briefly-dynamic tasks [3]. The major difference between our work and existing work in task and motion planning is that we focus on learning *mechanisms* that are sequences of basis manipulation operations and show significant improvement in complex manipulation tasks.

Our algorithm draws inspiration from contact-based modeling approaches in robot manipulation. In particular, various methods have been presented to make manipulation plans in the contact space, between rigid bodies as well as within robot hands [15, 16, 17, 18, 19, 20, 21, 22, 23]. Algorithms such as CMGMP [24, 25, 4] can automatically generate possible contact modes based on three primitive modes: fixed contact, separating, and sliding contact. In contrast to these methods that mostly focus on the characterization of contacts and planning with them, we consider learning *mechanisms* from contact modes in a single demonstration to solve new tasks efficiently.

Some mechanisms acquired by our model have been traditionally described as *tool-use* skills [26, 27, 28, 29]. In comparison to these works, we presented a novel framework for generating tool use trajectories with a planner based on contact sampling and showed that the learned mechanisms can be recombined with a general task and motion planner to solve more complex tasks. Recently, people have explored learning to predict tool-using trajectories or "object affordances" [30, 31, 32, 33, 34, 35], usually from a given set of demonstrations. By contrast, this paper focuses on generating trajectories with novel objects based only on a single demonstration, and combining the learned skills.

## 6 Limitations and Conclusions

Our mechanism learning framework faces two key challenges. First, obtaining detailed contact information from demonstrations can be difficult. This issue could potentially be addressed by integrating computer vision techniques capable of detecting contacts from videos. Second, the utilization of physical simulators for both learning and planning may not accurately replicate real-world physics, limiting the transferability of mechanisms to real robots. Future work should explore real-world fine-tuning to tackle this limitation.

In conclusion, this paper has introduced a novel framework that enables machines to *learn, reuse, and generalize* manipulation strategies (i.e., the *mechanisms*) from a single demonstration and subsequent self-play in a distribution of tasks. By characterizing each mechanism as a sequence of contact mode changes, the framework achieves notable generalization and compositionality. It generates successful trajectories for both novel object instances and categories under a briefly-dynamic setting, and allows for the recombination of learned mechanisms to solve more complex tasks. Our framework can also be flexibly extended by incorporating other basis operations such as compliance and forceful motion.

**Acknowledgement.** We thank anonymous reviewers for their valuable comments. This work is in part supported by ONR MURI N00014-16-1-2007, the Center for Brain, Minds, and Machines (CBMM, funded by NSF STC award CCF-1231216), NSF grant 2214177; AFOSR grant FA9550-22-1-0249; from ONR MURI grant N00014-22-1-2740; ARO grant W911NF-23-1-0034; the MIT-IBM Watson AI Lab; the MIT Quest for Intelligence; and the Boston Dynamics Artificial Intelligence Institute. Any opinions, findings, and conclusions or recommendations expressed in this material are those of the authors and do not necessarily reflect the views of our sponsors.

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

# Supplementary Material for
# Learning Reusable Manipulation Strategies

## Table of Contents

## A  Method Detail

### A.1  Planning Algorithm

Algorithm 2 shows the bi-level search algorithm we use. In the discrete search level, we enumerate both basis operations as well as mechanism operations. During the continuous parameter search phase, for basis operations instantiated from mechanisms, we use the mechanism-specific sampler rather than the generic sampler for continuous parameters.

---

**Algorithm 2** Bilevel Search Algorithm

---

1: **procedure** BILEVELSEARCH($s_0, g$, operator_schemas)
2:     $plan\_gen \leftarrow$ SymbolicSearch($s_0, g$, operator_schemas)       ▷ Explore discrete plans
3:     **for all** $plan \in plan\_gen$ **do**      ▷ For all candidate sequences of basis operations
4:         CONTINUOUSSEARCH($s_0, g, plan$)
5: **procedure** CONTINUOUSSEARCH($s_0, g, plan$)
6:     $grounded\_plan \leftarrow \emptyset$;  $s \leftarrow s_0$
7:     **for all** $op \in plan$ **do**
8:         **for all** $arg \in op.args$ **do**
9:             $arg \leftarrow$ InvokeSampler($op.sampler$)      ▷ Generate continuous parameters
10:        **if** CheckPrecondition($op, s$) **then**
11:           $s \leftarrow \mathcal{T}(s, op)$      ▷ Simulate the operator with sampled parameters.
12:           $grounded\_plan \leftarrow grounded\_plan \cup \{op\}$
13:        **else break**
14:     **if** IsGoalAchieved($grounded\_plan, g$) **then**
15:        **return** $grounded\_plan$      ▷ Return the first plan that achieves the goal
16:     **return** $empty$

---

### A.2  Briefly Dynamic Manipulation

The system can handle robot-object and object-object contact without assuming quasi-static motion. For example, when placing objects on surfaces, we consider subsequent pose changes: objects placed on inclined surfaces may slide down, and heavy objects placed on levers can alter the pose of the

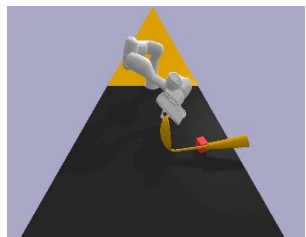 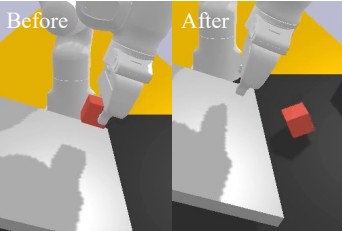 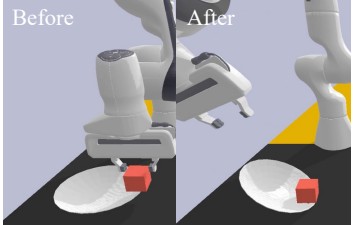

**Case 1:** non-rigid attachments between objects while moving.

**Case 2:** object pose change after placement due to physics.

**Case 3:** support object pose change after new objects being placed.

Figure 7: Illustration of three briefly-dynamic manipulation scenarios in the paper.

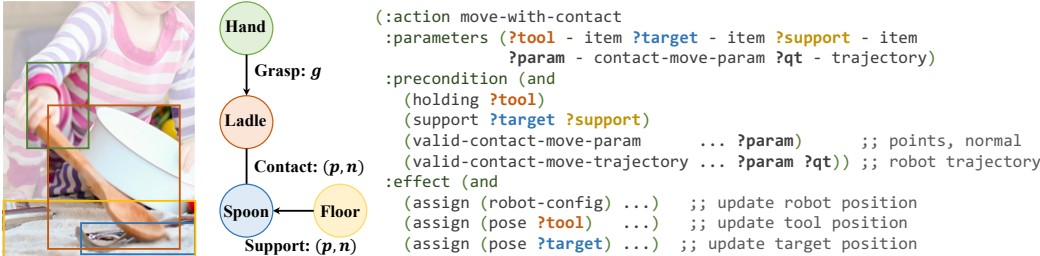

Figure 8: The modeling of the robot basis operation *move-with-contact* using a STRIPS-like syntax.

lever. Formally, we assume a *briefly-dynamic* manipulation setting, where the robot controller is position control-based, and manipulated objects may experience acceleration and velocity until they reach a stable configuration. Figure 7 illustrates a few examples of briefly-dynamic manipulation tasks handled by our sampler and planner.

### A.3  Basis Operations

In our manipulation context, each schema represents either a robot action that does not change the contact mode graph (e.g., moving the arm without object collisions) or a primitive action that modifies the contact mode (e.g., grasping an object). Table 1 shows the complete list of basis operators used in this paper.

The precondition and the effect of an action schema describe the contact relationships between objects before and after the execution. Fig. 8 showcases a concrete example. The action schema involves three objects: the object being held *?tool*, the target object that is in contact with the tool object *?target*, and the object that is currently supporting the target object *?support*. Additionally, there are two continuous parameters: *?param* specifies the contact between *?target* and *?tool*, including the contact surface and contact normal; *?qt* specifies the robot arm trajectory as a sequence of joint-space waypoints. This action updates the robot joint angles, and the poses of *?tool* and *?target*. Given the discrete and continuous parameters, we use a joint-space position controller to execute the actions and use the execution results to update the state variables.

We will present the implementation details for the samplers associated with each operator in the next section (Appendix A.4). At a high level, these samplers are designed to be very generic: for grasping, it randomly samples two parallel surfaces on objects; for object-object contact, it randomly samples two surfaces on the object and then transforms the object held by the robot so that two surface normals point to each other.

### A.4  Samplers for Generic Operations

Recall that there are three types of continuous variables to be sampled for the basis operators described in Table 1: grasping poses relative to an object (represented as $SE(3)$ poses of the end-effector relative

to the object), placement poses (represented as *SE*(3) poses in the support object frame), contacts between two objects (represented as the *SE*(3) pose of object 1 in the frame of object 2), and robot arm trajectories (represented as a sequence of arm trajectories). Here, we supplement the list of samplers we use to generate these continuous parameters. They are designed to be generic, relying solely on geometry and not specific object semantics (e.g., soup ladle grasping).

***Grasp (G).*** The grasp sampler, $G(\mathcal{O}, T_o)$, accepts the object's shape and current pose, $\mathcal{O}$ and $T_o$ respectively, and identifies two "parallel" surfaces on the object mesh, represented as $(p_1, n_1)$ and $(p_2, n_2)$, where $p_1$ and $p_2$ are two points and $n_1$ and $n_2$ are surface normals. The definition of being parallel is that: $(p_1 - p_2) \cdot n_1 = 1$ and $n_1 \cdot n_2 = -1$. It then computes a corresponding robot end-effector pose $T_{ee}$ such that $T_e e$ centered at the midpoint between $p_1$ and $p_2$, and $T_{ee}$ is perpendicular to $n_1$. It then checks the distance between two surfaces so that the parallel gripper can hold the object at $T_{ee}$. Finally, it checks the reachability of $T_{ee}$ using an inverse-kinematics solver.

***Placement (P).*** The placement position sampler, $P(\mathcal{O}_1, \mathcal{O}_2, T_{o2})$, considers the shapes of both the holding object, $\mathcal{O}_1$, and the target support object, $\mathcal{O}_2$, and the pose of $\mathcal{O}_2$. It randomly samples two surfaces, represented as $(p_1, n_1)$ and $(p_2, n_2)$, one on each object such that $n_2 \cdot (0, 0, 1)^T > 0.9$ (i.e., $n_2$ is close to the $+z$ direction). Next, it solves for a transform $T$ on $\mathcal{O}_1$ such that $T p_1 = T_{o2} p_2$ and $T n_1 = -T_{o2} n_2$ (essentially place $p_1$ on $\mathcal{O}_1$ at $p_2$ and pointing towards $n_2$).

***Object Contact (C).*** For both robot-object and object-object contact, the object contact sampler, $C(\mathcal{O}_1, \mathcal{O}_2, T_{o2}, \mathcal{O}_s, T_s)$ takes five arguments, including the current holding object $\mathcal{O}_1$ (or the robot gripper itself when not holding anything), the object to contact $\mathcal{O}_2$ and its current pose $T_{o2}$, and the object that supports $\mathcal{O}_2$: $\mathcal{O}_s$ and its pose $T_s$. It first randomly samples two surfaces, represented as $(p_1, n_1)$ and $(p_2, n_2)$ on $\mathcal{O}_1$ and $\mathcal{O}_2$ respectively. Since we do not consider pushing $\mathcal{O}_2$ "towards" the supporting object $\mathcal{O}_s$, we additionally require that $n_2$ is perpendicular to $n_s$, which is the direction of the support force from $\mathcal{O}_s$ to $\mathcal{O}_2$. Next, it solves for a transform $T$ on $\mathcal{O}_1$ such that $T p_1 = T_{o2} p_2$ and $T n_1 = -T_{o2} n_2$ (essentially place $p_1$ on $\mathcal{O}_1$ at $p_2$ and pointing towards $n_2$ to excert force).

***Trajectory (T).*** For grasping and placement trajectories, the trajectory sampler, $T(T_{init}, T_{target})$, considers the initial and target end-effector pose of the robot gripper. It first uses an inverse kinematic solver to solve for two robot configurations at the designated end-effector pose $q_{init}$ and $q_{target}$. Next, we compute a collision-free trajectory (except for collisions with the object being held and the object to contact) using a Bidirectional Rapidly-exploring Random Tree (BiRRT) algorithm.

For move-with-contact trajectories, the trajectory sampler, $T(T_{init}, p_1, n_1, p_2, n_2)$, accepts the initial configuration of the robot, $g_{init}$, and the contact surfaces on the two objects sampled using the object contact sampler $C$: $(p_1, n_1)$ and $(p_2, n_2)$. It proceeds to randomly sample a "push" distance, $d$, along the contact normal direction, $n_1$, from a uniform distribution in the range $[0.05, 0.25]$ meters. Subsequently, the sampler generates the arm trajectory by invoking the BiRRT algorithm to follow a set of waypoints corresponding to a linear Cartesian-space motion along $n_1$ by distance $d$.

# B  Experiment Detail

## B.1  Mechanism Learning Setup

Our evaluation encompasses six distinct mechanisms, grouped into two categories: the first four tasks assess "tool-use."

*(Edge)* pushing objects to the edge of a table for pickup. There are four object models used in this mechanism: plate, calculator, caliper, and document.

*(Hook)* using tools to reach for distant objects. There are five objects that can be used as the "hook:" wooden L-shape stick, soup ladle, hammer, spoon, and caliper.

*(Lever)* flipping objects using heavy objects as levers. There are four "heavy" objects that can be used to flip the plate: box, spoon, dipper, and walnut.

*(Poking)* using tools to poke objects out of a tunnel. There are three object models that can be used as the "poking" tool: wooden stick, spatula, and spoon.

The remaining two tasks fall under the "reasoning about stability" category.

*(Center-of-Mass)* achieving stable object placement on another object. There are three object models to be placed on the small block: plate, calculator, and document.

*(Slope-and-Blocker)* using objects as blockers to prevent objects from falling off inclined surfaces. There are three object models that can be used as the blocker: wooden stick, wooden L-shape stick, and spoon.

For each environment, we first manually defined a canonical pose for each object such that the mechanism is feasible. Next, for each training and testing instance, we randomly apply small translations (a uniform distribution within $\pm$ 5 centimeters) and small rotations (uniform within $\pm$ 15 degrees) to the canonical pose of each movable object.

## B.2  Sampler Learning

Taking a closer look at the importance of sampler learning, Fig. 9 illustrates a breakdown of the number of samples required for the "hook use" mechanism using our planning algorithm, with the generic sampler and with the learned sampler. Fig. 3b shows the inferred macro definition for this mechanism, and here we count the number of samples produced by each individual sampler. In this case, most of the samplers are produced to generate candidate grasping poses of the tool and possible contacts between the tool and the target (i.e., how to reach the tool).

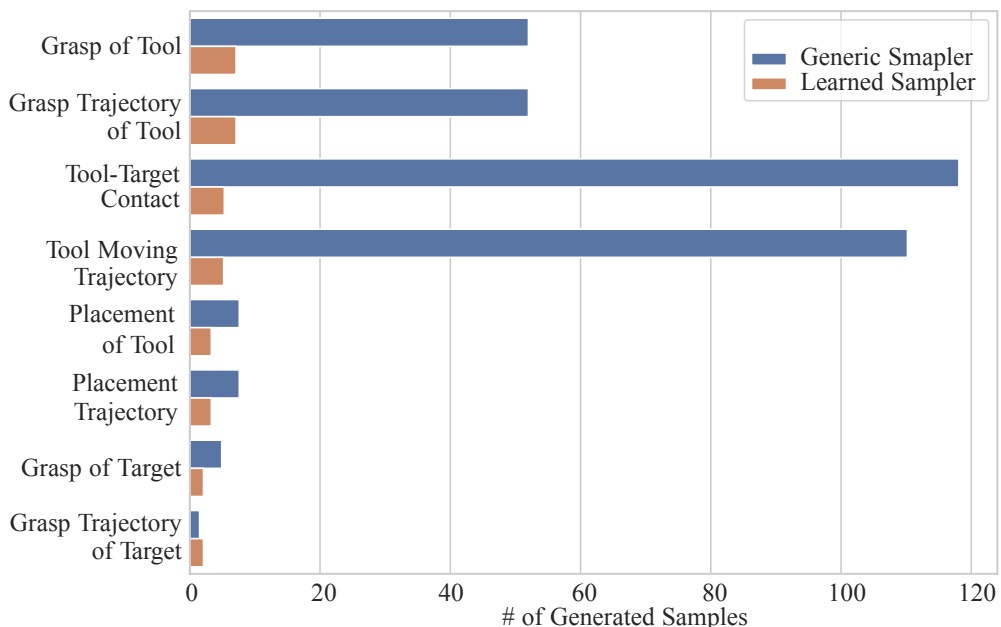

Figure 9: Breakdown of samples produced by different samplers for the hook-using task.

## B.3 Physical Robot Deployment

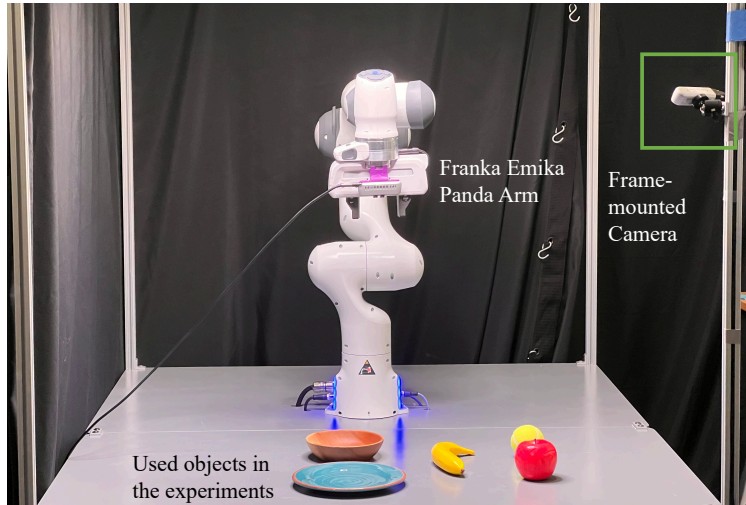

Figure 10: Our real-world setup.

| Task | push-to-edge(plate) | hook-use(banana, {ball, apple}) |
|------|---------------------|--------------------------------|
| Succ. Rate | 80% | 70% |

Table 4: Success rate of our physical robot setup on two representative mechanisms.

Our real-world setup, shown in Fig. 10, contains a Franka Emika Panda robot arm with a parallel gripper, mounted on a table. We also have an Intel Realsense D435 camera mounted on the table frame, pointing 45 degrees down. Our vision pipeline contains six steps: first, based on the calibrated camera intrinsics and extrinsics, we reconstruct a partial point cloud for the scene. Second, we crop the scene to exclude volumes outside the table (e.g., background drops, etc.) Third, we use a RANSAC-based algorithm to estimate the table plane, and extract all object point clouds on top of the table. Fourth, we use Mask-RCNN to detect all objects in the RGB space, and extract the corresponding point clouds. For objects that are not detected by the MaskRCNN, we first use DBScan to cluster their point clouds, and then run the SegmentAnything model to extract their segmentations in 2D, and subsequently the point clouds. Finally, we perform object completion by projecting and extruding the bottom surface for all detected objects down to the table.

Next, we import the reconstructed object models into our physical simulator PyBullet, and directly deploy our planning algorithm with the learned mechanisms to compute plans. Since we use the same robot model in simulation, we can directly execute the planned robot trajectories in the real world. In practice, we execute in a closed-loop manner. If the grasp fails (which can be detected by the gripper sensor), we move the arm to its neutral position and replan. After each placement action and pushing action, we use the vision pipeline to obtain an updated world state and replan.

We demonstrate and evaluate the system on two example tasks: pushing plates to the edge and hook-using for distant objects. We use the same set of objects on the table but with different initial configurations. We repeat each experiment 10 times and report the success rate. Table 4 summarizes the result. The visualization of our vision pipeline and the robot videos can be found on our website.

The push-to-edge task is relatively easier given the learned mechanism. The only two failure cases in our experiment were triggered by the robot pushing the plate too far and the plate falling off the table. The hook-using task is more challenging. There are two main failure modes we observe in the experiments. First, the planner favors grasping objects (e.g., the banana "hook") at the tip, which is a very unstable grasp—sometimes causing the executor to fail. Also, sometimes the planner generates

wrong object motion for the objects because 3D of reconstruction errors: the planner is using a "hallucinated" part of the object to push the distant object—currently, we cannot recover from this type of failure. Finally, sometimes the sampled grasp is invalid, again because of the errors in object perception. In this case, our close-loop execution improves the performance.

