# OpenReview forum: "Learning Reusable Manipulation Strategies"
_robot-learning.org/CoRL/2023/Conference — CoRL 2023 Poster_

### Official Review · Reviewer_5d1G · 2023-07-07

**Confidence:** 3
**Originality:** Good
**Technical Quality:** Very Good
**Clarity Of Presentation:** Very Good
**Impact:** 2

**Recommendation:**

Weak Accept: I recommend accepting the paper, but will not argue for my recommendation if the majority of other reviewers have a different opinion.

**Review:**

- One major weakness which is also discussed in the limitations section by the authors are the missing real robot experiments.

- I understand that the presented concept is sampling based and can be integrated into sampling-based task planners. I would like to see a clarifying statement on why a comparison to existing approaches is difficult, since in the exerperimental section the presented method has only been compared to a baseline approach.

- As also discussed by the authors uncertainty and imprecise demonstrations are a major issue with the approach. However, I also believe those topics can be tackled by follow-up work. Additionally, the aforementioned difficulties are shared with other planning approaches and are not specific for the presented approach.

- Line 154: they present -> they are present

**Quality Of The Limitations Section:**

Limitations are addressed clearly

**Questions For Rebuttal:**

- Why is a comparison with other state-of-the-art approaches not feasible? Why did you choose your own baseline over existing approaches?

**Robotics Focus:**

Highly relevant to robotics but no hardware experiments

**Summary Of Paper:**

The authors present an approach for learning manipulation mechanisms. A mechanism is a reusable sequence of "manipulation primitives" that comes with an individual sampling strategy for each primitive. While the primitives seem to be predefined, the learned manipulation mechanisms can be reused in other learning setups. Individual primitives and mechanisms are parametrized such that they are not limited to single tools but can also be generalized to setups which are different from the single demonstration that was used to learn them.

**Summary Of Recommendation:**

I chose my rating mainly because I did not find any major flaws in the paper. I don't think my major concerns and questions about the approach in general have to be tackled in this paper, but can also be follow-up work.

---

### Official Review · Reviewer_KXeh · 2023-07-18

**Confidence:** 3
**Originality:** Very Good
**Technical Quality:** Good
**Clarity Of Presentation:** Good
**Impact:** 3

**Recommendation:**

Weak Accept: I recommend accepting the paper, but will not argue for my recommendation if the majority of other reviewers have a different opinion.

**Review:**

## Strengths and Weaknesses
### Strengths

 - An automated method for learning reusable strategies from limited number of demonstrations
 - An answer to an important question of how to build reusable manipulation skills
 - A well written paper

### Weaknesses
 - Unclear scalability to physical systems
 - Convoluted algorithm that does not necessarily scale to dexterous tasks

**Quality Of The Limitations Section:**

Limitations are addressed clearly

**Questions For Rebuttal:**

- Why was this approach not tried on a physical system, substituting some of the skill learning techniques with more classical approaches and/or approaches that rely less on online learning can alleviate a lot of the constraints
- How can this approach scale to more complex dexterous tasks, such as insertion with tight tolerances, drawer opening etc

**Robotics Focus:**

Relevant but unlikely to deploy to hardware in near future

**Summary Of Paper:**

This work introduces a novel method for learning reusable manipulation strategies. The core functionality of the proposed approach is the ability to sequence single demonstrations into smaller, core skills that can be sequenced together. The individual skills are then learnt through trial and error and by using self-play. Overall, the paper is well written and clearly discusses both pros and cons of the proposed approach. The evaluation, conducted in simulated environments, is extensive and well done. My only big concern, also acknowledged in the limitations section, is that this method is not straightforwardly generalised to physical systems. Having said that, I find this work as an excellent paper but question its fit for CoRL.

**Summary Of Recommendation:**

Overall, I like this approach but I am not convinced it is a great fit for the CoRL community. Having said that, it seems like a well executed paper and as a result I recommend a weak reject with mild confidence purely on the basis of its fit to the CoRL community.

---

### Official Review · Reviewer_sZoG · 2023-07-21

**Confidence:** 3
**Originality:** Very Good
**Technical Quality:** Good
**Clarity Of Presentation:** Very Good
**Impact:** 4

**Recommendation:**

Strong Accept: I recommend accepting the paper and will argue for my recommendation even if other reviewers hold a different opinion.

**Review:**

Strengths:

- The paper presents a new framework for learning a useful manipulation strategy composed of multiple basis operations (or lower-level skills) that utilises the change in a contact mode graph to understand the transition between basis operations.
- The idea I believe is unique and is framed in a way such that it is useful for long-horizon task planning where there are also continuous variables (task and motion planning).
- The authors propose a method for learning new mechanisms from a single demonstration and then utilising self-play to improve the generalisation of the mechanism to a known set of objects.
- The paper is generally well written with ideas clearly and logically presented although there are points where refinement could help to avoid confusion. For example, the delineation between skills, tricks, mechanisms, strategies, basis operations etc. could be made clearer and earlier.

Weaknesses:

- The authors fail to compare their method against any competitive or state-of-the-art baseline. An option here might be a hierarchical RL policy seeing as there are a fixed set of objects and all experiments are completed in simulation.
- There is mention that this method is able to handle briefly-dynamic tasks, however, there is no proof of this in the main body of the paper and seeing as planning and execution is completed in simulation it is difficult to see how this would remain true if transferred to real-world environments.
- The limitations of devising a manipulation strategy as a mechanism are not explored nor are the requirements for the demonstration.

**Quality Of The Limitations Section:**

Additional details required

**Questions For Rebuttal:**

What is the limit to the number of basis operations that can be composed in a single mechanism? Are there downsides of composing long strategies?


Only 100 samples are used to train each scoring classifier for predicting successful samples in simulation with limited objects and starting configurations. This seems remarkably few. Scaling the scoring classifiers to real-world environments appears like it will become prohibitively expensive. Do you believe the method, on top of the other obvious sim-2-real issues, will be able to scale in this way to transition to real-world setups?

**Robotics Focus:**

Highly relevant to robotics but no hardware experiments

**Summary Of Paper:**

The submitted paper presents a new framework for learning reusable manipulation "mechanisms". These mechanisms can be learnt from a single demonstration and then additional self-play. A mechanism learns the sequence of contact mode changes between the robot, objects and support surfaces. Planning with mechanisms assists with efficiently creating long-horizon plans.

**Summary Of Recommendation:**

The authors have presented a paper that introduces a framework for learning "mechanisms" that are composed of lower-level basis operations. The idea leverages the sequence of contact mode changes in a demonstration to efficiently learn. Overall the work is well presented although is missing some discussion on the limitations of the proposed method and the requirements for the demonstration.

---

### Official Review · Reviewer_W6qd · 2023-07-22

**Confidence:** 2
**Originality:** Fair
**Technical Quality:** Good
**Clarity Of Presentation:** Good
**Impact:** 3

**Recommendation:**

Weak Reject: I recommend rejecting the paper, but will not argue for my recommendation if the majority of other reviewers have a different opinion.

**Review:**

The paper presents a problem that is relevant for the robot learning community - being sample efficient and being able to generalise well in long-horizon tasks.

Strengths:
* the authors choose to look at learning from demonstration as a hybrid system which ultimately means representing some of the information encoded in the user demonstrations as symbols. As a result they can utilise off-the-shelf proven planning algorithms and planning languages which in turn allows them to be very efficient and to generalise to novel skills, even non-demonstrated ones.

Weaknesses:
* the method is heavily dependant on having state information from the simulator about contacts, in order to be able to decompose a single demonstration into its subparts.
* the paper is generally hard to read and introduces a lot of it's own naming and terms that are not clearly explained and examplified - i.e. manipulation skills are mechanisms, briefly-dynamic, learned specialised samplers (I'm still not sure what are the samplers even after reading the paper a few times.)
* all experiments are solely done in simulation, partly because of the method is too dependent on having full state observability. It is unclear whether and how much the method would succeed in scenarios where that's not the case - i.e. the real world with physical robots, noisy sensors and very-hard-to-estimate contacts purely from vision.

**Quality Of The Limitations Section:**

Additional details required

**Questions For Rebuttal:**

* how much effort do you think it will be to have this method run on a real robot? Would it still be able to support 1-shot learning or maybe it would need more demonstrations per task?
* what happens if there are distractor objects in the scene - is the method capable of figuring out what matters in a scene and what doesn't with its current STRIPS formulation?
* fixing the set of basis operators ultimately means that the method won't be able to support certain skills - what behaviours can't the robot learn and do?

**Robotics Focus:**

Relevant but unlikely to deploy to hardware in near future

**Summary Of Paper:**

The authors present a framework for learning manipulation skills, which they refer to as mechanisms, and are composed of sequences of defined basis manipulation operations. The problem of manipulation is looked at as a STRIPS planning problem. As a result, even a single user demonstration of a task can be decomposed into parametrised manipulation operations and then repeated. The author present results from their method in simulation on a series of manipulation tasks of varying complexity.

**Summary Of Recommendation:**

I recommend weak reject but would be willing to change my recommendation during the rebuttal period.

---

### Author Response · Authors · 2023-08-11
**Real-World Setup and Experiments**

We thank all reviewers for their thoughtful comments and helpful suggestions. We have responded to your individual questions. Since most reviewers have asked about the deployment of our algorithms in the real world, we iilustrate a possible integration of our planning algorithm and a vision pipeline to deploy the system to real robots. Our implementation does not assume groundtruth vision information such as object models and poses. By contrast, we use RGBD cameras and pretrained object segmentation models to segment objects from observed point clouds. Then, we use our planner with an internal simulator to compute possible plans. Finally, we execute the actions with a Franka Emika Panda robot arm with parallel gripper mounted on the table. We include detailed descriptions, visualizations, and results below.

**Note:** Since OpenReview does not allow authors to upload a general "rebuttal" with files, we have uploaded the visualization images and videos as attachments to our individual responses to reviewers.

**Implementation Details**

Our real-world set up contains a Franka Emika Panda robot arm with a parallel gripper, mounted on a table. We also have a Intel Realsense D435 camera mounted on the table frame, pointing 45 degrees down. Our vision pipeline has six steps: first, based on the calibrated camera intrinsics and extrinsics, we reconstruct a partial point cloud for the scene. Second, we crop the scene to exclude volumes outside the table (e.g., background drops, etc.) Third, we use a RANSAC-based algorithm to estimate the table plane, and extract all object pointclouds on top of the table. Fourth, we use Mask-RCNN to detect all objects in the RGB space, and extract the corresponding point clouds. For objects that are not detected by the MaskRCNN, we first use DBScan to cluster their point clouds, and then run SegmentAnything model to extract their segmentations in 2D, and subsequently the point clouds. Finally, we perform object completion by projecting and extruding the bottom surface for all detected objects down to the table. We provide visualizations of the reconstructed point clouds in the attached zip file.

Next, we import the reconstructed object models into our physical simulator PyBullet, and directly deploy our planning algorithm with the learned mechanisms to compute plans. Since we use the same robot model in simulation, we can directly execute the planned robot trajectories in the real world. In practice, we execute in a closed loop manner. If the grasp fails (which can be detected by the gripper sensor), we move to arm to its neutral position, gather a new observation, and replan. In addition, after each placement or pushing action, we use the vision pipeline to obtain an updated world state and replan.

**Results**

We demonstrate and evaluate the system on two example tasks: pushing plates to the edge and hook-using for distant objects. We use the same set of objects on the table but with different intial configurations. We repeat each experiment for 10 times and report the success rate.

| Task       | push-to-edge(plate) | hook-use(banana, {ball, apple}) |
| --------   | -------- | -------- |
| Succ. Rate | 80%      | 70%      |

The push-to-edge task is relatively easier given the learned mechanism. The only two failure cases in our experiment were triggered by the robot pushing the plate too far and the plate falling off the table. The hook-using task is more challenging. There are two main failure modes we observe in the experiments. First, the planner favors grasping objects (e.g., the banana "hook") at the tip, which is a very unstable grasp---sometimes causing the executor to fail. Also, sometimes the planner generates wrong object motion for the objects because 3D of reconstruction errors: the planner is using a "hallucinated" part of the object to push the distant object---currently, we cannot recover from this type of failure. Finally, sometimes the sampled grasp is invalid, again because of the errors in object perception. In this case, our close-loop execution improves the performance.

---

### Decision · Program_Chairs · 2023-08-30

**Decision:**

Accept (Poster)

**Comment:**

This work presents an approach to learn reusable skills, referred to as "mechanisms". These mechanisms learn sequences of contact mode changes between the robot and objects based on a single virtual demonstration and subsequent self-play. While the evaluations in the manuscript were only performed in simulation, the authors provide real world experiments during the rebuttal period. Unfortunately, the approach still requires an initial virtual demonstration even for the real world experiments, which is not ideal but I it should not be a disqualifying limitation.

The authors generally did a good job in the rebuttal and I believe given the revisions described in the rebuttal that this work can be accepted for CoRL 23.